# Fast and Stable Continual Test-Time Adaptation via Masked Modeling and Momentum-Guided Updates

## Abstract

Continual test-time adaptation (CTTA) aims to maintain model accuracy under non-stationary distribution shifts when source training data are unavailable. Existing methods using pseudo labels struggle to balance rapid adaptation with knowledge retention, often requiring multiple forward passes, which makes them impractical for deployment under strict latency constraints. We present MiDEA (**M**asked-**i**mage modeling with **D**ual-**E**MA **A**daptation), a decoder-free framework with one teacher and two student forward passes per batch, combining global two-view consistency, masked local alignment, and layer-wise dual-rate EMA. MiDEA maintains a teacher-student architecture that measures distribution gaps at both global image and local levels. Globally, it aligns clean teacher predictions with strongly augmented student views and locally, it matches teacher patch representations to student masked embeddings. The teacher updates via dual-rate EMA: attention layers adapt rapidly while MLP weights drift slowly, reducing drift during continual updates. On ViT-Base, MiDEA achieves 38.1% ImageNet-C error, 18 points below frozen models and 5 below previous CTTA state-of-the-art, while running 3× faster than multi-pass methods and maintaining accuracy at batch size 1.

## 1 Introduction

Real-world deployment of deep neural networks frequently encounters *test-time distribution shifts*, which can cause substantial performance degradation. Despite advances in supervised and unsupervised representation learning Dosovitskiy et al. (2021); He et al. (2022); Caron et al. (2021), trained models typically suffer severe accuracy drops when faced with unexpected data variations at inference time Hendrycks & Dietterich (2019). Autonomous vehicles and robotic systems must remain robust under changing weather, illumination, and environmental context (see examples in Figure 1).

*Test-time adaptation* (TTA) can help under a single, approximately stationary shift, yet operational conditions evolve over time as dawn becomes daylight and clear skies turn to rain. This motivates the more challenging setting of *continual test-time adaptation* (CTTA) Wang et al. (2022).

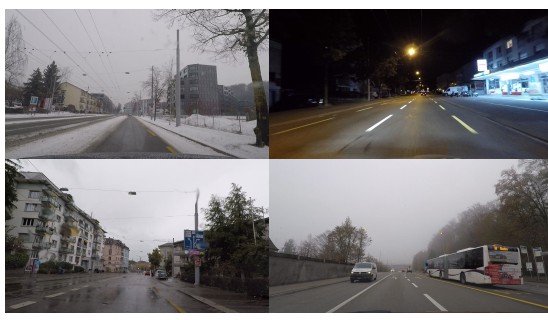

Figure 1: Real-world environments frequently introduce continuous distribution shifts (weather changes, lighting, noise, blur), motivating continual test-time adaptation (CTTA).

CTTA introduces two central difficulties: *catastrophic forgetting* and *error accumulation*. Methods based on pseudo labels and teacher–student training, including CoTTA Wang et al. (2022),

Continual-MAE Liu et al. (2024a), and RMT Doebler et al. (2023), often require multiple forward passes, auxiliary decoders, or sizable memory buffers, which hinders deployment in resource-constrained applications. This raises a fundamental question: *Can we achieve state-of-the-art continual robustness without heavy computational overhead?*

**Our Approach.** MiDEA (Masked-image modeling with Dual-EMA Adaptation) is a decoder-free CTTA method for Vision Transformers that uses a single teacher pass per batch. The pipeline combines three elements: (i) a standard global two-view consistency term that aligns a clean teacher view with a strongly augmented student view, (ii) a masked local self-distillation loss that matches internal patch embeddings on masked regions without any reconstruction decoder, providing spatial regularization, and (iii) a layer-wise dual-rate EMA update that adapts attention blocks faster than MLP blocks to balance plasticity and stability.

**Empirical Results.** On ImageNet-C Hendrycks & Dietterich (2019), MiDEA attains 38.1% error, improving by 18 points over non-adaptive baselines and by 5 points over prior ViT-B results, while processing 454 images/s on an RTX 3080, over 3x faster than recent CTTA methods Wang et al. (2022); Liu et al. (2024a); Doebler et al. (2023). Similar gains are observed on CIFAR-10-C (17.6 pp) and CIFAR-100-C (11.4 pp). Accuracy remains stable at batch size 1, which supports latency-constrained deployments.

**Contributions.** (1) A minimal, source-free CTTA design for ViTs that is single-teacher-pass and decoder-free. (2) A masked local alignment objective that complements global consistency without auxiliary decoders. (3) A layer-wise dual-rate EMA update that improves stability over single-rate baselines. (4) A practical accuracy–throughput profile with stable performance even at batch size 1.

## 2 RELATED WORK

### 2.1 SELF-SUPERVISED LEARNING AND MASKED IMAGE MODELING

Self-supervised learning (SSL) has evolved from simple proxy tasks, such as rotation prediction Gidaris et al. (2018) and image colorization Larsson et al. (2017), to more advanced contrastive methods that learn representations by drawing semantically similar samples closer while pushing dissimilar ones apart Chen et al. (2020a); He et al. (2020); Chen et al. (2020c); Grill et al. (2020); Caron et al. (2021); Chen & He (2021); Chen et al. (2020b); Assran et al. (2023).

A particularly relevant branch of SSL is masked image modeling (MIM), pioneered by Masked Autoencoders (MAE) He et al. (2022). MAE randomly masks 75% of image patches, encodes only visible patches, and uses a lightweight decoder to reconstruct the complete RGB image. Subsequent work has shown that reconstructing engineered features, such as Histogram-of-Oriented-Gradients (HOG) Wei et al. (2022), instead of raw pixels can improve performance. SimMIM Xie et al. (2022) simplifies this pipeline by passing both masked and unmasked patches through the encoder, whereas hybrid approaches, such as iBOT Zhou et al. (2022), combine masked modeling with contrastive objectives.

The inherent label-free learning signal in SSL methods makes them naturally suited for continual test-time adaptation. Recent CTTA methods have successfully integrated SSL objectives into the test-time loop: Contrastive TTA Chen et al. (2022) adds InfoNCE-style contrastive loss between augmented views, while Continual-MAE (ADMA) Liu et al. (2024a) adapts MAE's masking strategy for online adaptation. However, ADMA requires multiple forward passes for patch selection, increasing computational cost.

### 2.2 TEST-TIME ADAPTATION

Test-time adaptation (TTA) addresses the challenge of adapting pre-trained models to new, stationary target domains using only unlabeled test data. Early approaches like AdaBN Li et al. (2016) recompute batch normalization statistics, while Tent Wang et al. (2021) performs entropy minimization by updating only BatchNorm parameters.

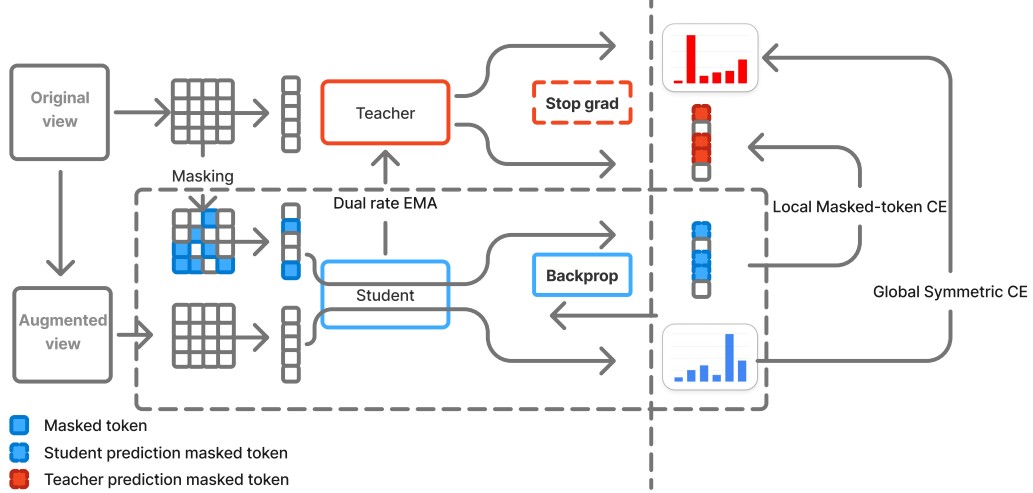

Figure 2: MiDEA's **holistic update loop**. The **student** processes the *original view* once with a random patch mask (local branch) and the *augmented view* without masking (global branch). **(1) Global two-view loss:** a *symmetric* cross-entropy aligns the full-image prediction vectors of teacher and student across the two views. **(2) Local same-view loss:** a decoder-free cross-entropy matches teacher and student *patch embeddings* only on the tokens that were masked, supplying fine-grained spatial guidance. A single backpropagation step updates the student, after which **(3) Dual-rate EMA** moves the **teacher** towards the new student weights, using a fast EMA for attention layers, and a slow one for Conv/MLP. This helps prevent drifting during continual updates without requiring extra passes. Thus, MiDEA couples *global+local* self-distillation with a layer-aware EMA in one decoder-free pass per view.

More sophisticated methods include SHOT Liang et al. (2020), which requires specialized source training with label smoothing, and TTT Sun et al. (2020), which incorporates rotation prediction as an auxiliary task. Recent TTA methods have explored entropy-based self-training Niu et al. (2022), contrastive objectives Liu et al. (2022), and parameter-efficient approaches like visual prompting Gao et al. (2022), and lightweight meta-network adaptation Song et al. (2023). While effective for static distribution shifts, these methods struggle when test distributions evolve continuously, motivating the development of continual test-time adaptation.

## 2.3 CONTINUAL TEST-TIME ADAPTATION

Continual test-time adaptation (CTTA) extends TTA to handle non-stationary test streams where distributions shift continuously over time. This setting introduces two critical challenges: *catastrophic forgetting*, where models lose previously learned knowledge when adapting to new data, and *error accumulation*, where adaptation mistakes compound over successive updates.

**Teacher-Student Frameworks.** The most recent CTTA methods employ teacher-student architectures, where the student adapts to new data while a slowly updated exponential moving average (EMA) teacher provides stability. CoTTA Wang et al. (2022) generates stable pseudo-labels by averaging teacher predictions over 32 random augmentations and periodically resets teacher weights to source values. VIDA Liu et al. (2024b) introduces LoRA adapter modules that can be updated online; however, it still requires 10 teacher passes per batch and necessitates source data for adapter pre-training. Although ADMA Liu et al. (2024a) does not explicitly use a teacher-student structure, it replaces the teacher's stabilization role with masked-image modeling. Still, it incurs overhead due to multiple forward passes and relies on reconstructing handcrafted features. **In contrast, MiDEA explicitly uses local semantics by directly aligning masked patch embeddings between teacher and student, eliminating the need for a reconstruction decoder.**

**Memory and Sample-Selection Methods.** Several other methods explicitly utilize memory banks or queues, such as RMT Doebler et al. (2023), RoTTA Yuan et al. (2023), and AR-TTA Sójka et al. (2023). Sample-selection approaches focus adaptation efforts on identifying reliable versus unreliable test samples, adjusting their update strategies accordingly. Examples include FACTTA Wu & Zhuang (2024), CCoTTA Shi et al. (2024), and Yang *et al.* Yang et al. (2024b;a). Such reliability-based strategies may also be complementary to our method Mounsaveng et al. (2024). **Parameter-Efficient Approaches.** Recent work explores minimal parameter updates through visual prompts. VDP Gan et al. (2023) learns domain-specific and domain-agnostic pixel-level prompts through teacher-student EMA, whereas DePT Gao et al. (2022) maintains online memory banks for pseudo-labeling, which introduces additional memory costs.

**Relation to Robust Mean Teacher (RMT).** RMT Doebler et al. (2023) also uses **two-view** teacher–student consistency. MiDEA differs by targeting ViTs with a *single teacher pass*, no replay/ensembling/contrastive memory, and a *decoder-free* masked local alignment plus *dual-rate EMA*. This yields a simpler, deployment-oriented recipe under strict efficiency constraints.

## 3 METHOD

### 3.1 PROBLEM SETUP

Let $f_{\theta_0}$ be a classifier with initial parameters $\theta_0$ trained on labeled source data $(X_S, Y_S)$, where $X_S$ and $Y_S$ denote the source domain inputs and labels, respectively. At test time, the model observes a continuous stream $\{x_t\}_{t=1}^{\infty}$ whose distribution $p_t(x)$ drifts continuously and unpredictably. Crucially, source samples are unavailable during adaptation, and each target sample is observed only once. After making a prediction $f_{\theta_t}(x_t)$, the model applies at most one gradient step to obtain updated parameters $\theta_{t+1}$. The objective is to minimize cumulative prediction error while avoiding catastrophic forgetting of the original source domain knowledge. This challenging setting, known as **continual test-time adaptation (CTTA)**, requires methods that can rapidly adapt to new conditions while maintaining knowledge of past distributions.

### 3.2 MIDEA OVERVIEW

MiDEA addresses CTTA through three synergistic innovations: (1) **global** image-level consistency using symmetric cross-entropy between clean and augmented views, (2) **local** patch-level alignment via masked self-distillation without reconstruction decoders, and (3) a **dual-rate EMA** mechanism that enables attention layers to adapt faster than MLP layers, balancing plasticity with stability. Our method requires two student and one teacher forward passes per batch (three total) (Figure 2).

### 3.3 GLOBAL TWO-VIEW CONSISTENCY

To establish robust image-level semantic alignment, we create two global crops from each input image:

$$x_u \quad \text{(clean, unaugmented)}, \tag{1}$$
$$x_v = \mathcal{T}(x) \quad \text{(strongly augmented)}, \tag{2}$$

where $\mathcal{T}(\cdot)$ represents strong data augmentation, including random affine and color jittering.

The teacher processes the clean image $x_u$ through a single forward pass to produce class predictions $q_u^* \in \mathbb{R}^C$. Simultaneously, the student processes the augmented view $x_v$ to generate corresponding predictions $p_v$. We enforce consistency between these predictions using a symmetric cross-entropy (CE) loss:

$$\mathcal{L}_{\text{cons}} = \frac{1}{2} \left[ \text{CE}(q_u^*, p_v) + \text{CE}(p_v, q_u^*) \right]. \tag{3}$$

The clean teacher view provides a stable semantic reference, while the student must match predictions under strong augmentation; symmetric cross-entropy mitigates pseudo-label noise Doebler et al. (2023), which is crucial in continual adaptation. This formulation is conceptually related to the two-view consistency in Robust Mean Teacher (RMT) Doebler et al. (2023), though MiDEA does

not adopt RMT's replay buffers, contrastive objectives, or ensembling, focusing instead on a minimal, single-teacher-pass design for ViTs. Unlike previous methods that require multiple augmented views and repeated teacher inferences, our approach achieves effective global alignment with one teacher and two student forward passes per batch.

### 3.4 LOCAL MASKED SELF-DISTILLATION

While global consistency captures image-level semantics, it ignores spatial structure and fine-grained details. To address this limitation, we introduce a complementary **local** loss that operates on patch embeddings, leveraging the network's internal representations rather than reconstructing pixels or handcrafted features.

We apply a binary mask to the patch tokens of the clean image $x_u$, where $N$ is the total number of patches and $m \subset \{1, \ldots, N\}$ denotes the set of masked patch indices with masking ratio $r = 0.5$. The student re-encodes this masked image to produce patch embeddings $z_{u,i}$ for each spatial location $i$. In parallel, the teacher processes the complete unmasked image to generate reference embeddings $z_{u,i}^*$.

Since both networks operate on the same spatial grid, we can directly compare embeddings at corresponding locations. We apply supervision only on the masked patches, where the student lacks direct visual information:

$$\mathcal{L}_{\text{mim}} = -\sum_{i \in m} z_{u,i}^* \cdot \log z_{u,i} = \sum_{i \in m} \text{CE}(z_{u,i}^*, z_{u,i}). \tag{4}$$

This masked self-distillation encourages the student to learn fine-grained spatial structure precisely where it lacks information, turning local alignment into a non-trivial prediction task that builds context-aware features rather than copying visible tokens. Our ablation analysis demonstrates that this decoder-free token-to-token alignment contributes approximately 2.3 percentage points in accuracy improvement.

**Evaluation protocol.** Unless otherwise specified, all reported results are obtained from the EMA teacher model applied to the clean, unaugmented view at test time.

### 3.5 LAYER-WISE DUAL-RATE EMA

Standard EMA methods apply the same update rate to all network parameters, which creates a one-size-fits-all plasticity level. This can lead to two issues: either the network forgets information too quickly or it adapts too slowly. Different components of the network have specific functions; for instance, attention layers are designed to capture contextual relationships and may benefit from adapting more quickly to new visual patterns. In contrast, MLP layers encode fundamental representations that should change more conservatively.

We address this challenge by maintaining separate EMA rates for different layer types. Let $\theta_\ell^{(t)}$ denote the student parameters for layer $\ell$ at time $t$, and $\theta_\ell^{*(t)}$ denote the corresponding teacher parameters. The teacher's update rule is:

$$\theta_\ell^{*(t+1)} \leftarrow \alpha_\ell \theta_\ell^{*(t)} + (1 - \alpha_\ell)\theta_\ell^{(t+1)},$$
$$\alpha_\ell = \begin{cases} \alpha_{\text{weight}} & \text{if } \ell \text{ is MLP/conv layer,} \\ \alpha_{\text{attn}} & \text{if } \ell \text{ is attention/projection layer,} \end{cases} \tag{5}$$

where $\alpha_{\text{attn}} < \alpha_{\text{weight}}$, allowing attention layers to update more rapidly.

This dual-rate schedule enables fast-adapting attention mechanisms to specialize to new domains while slow-updating MLP and convolutional layers preserve long-term semantic knowledge. The result is a single set of parameters that maintains both recent adaptations and historical knowledge. Our ablation studies show this approach recovers approximately 2 percentage points over the best single-rate baseline, suggesting that explicitly separating plastic and stable components can be more effective than tuning a single global EMA rate. This separation improves stability by keeping attention layers responsive and MLP/conv layers conservative, reducing drift during continual updates.

### 3.6 COMBINED OBJECTIVE AND UPDATE PROCEDURE

The final loss function combines our global and local consistency terms:

$$\mathcal{L} = \lambda_{\text{mim}}\mathcal{L}_{\text{mim}} + \mathcal{L}_{\text{cons}}, \tag{6}$$

where $\lambda_{\text{mim}}$ controls the relative importance of local patch-level alignment. Unless otherwise noted, we use a fixed configuration *per dataset* (learning rate, EMA rates, $\lambda_{\text{mim}}$, masking ratio). Section 4.1 details both tuned settings and a *universal* configuration shared across datasets.

---

**Algorithm 1** MiDEA Single-Batch Update

---

1: **Input:** Clean batch $x_u$, mask ratio $r$
2: $x_v \leftarrow \text{StrongAugment}(x_u)$  ▷ Create augmented view
3: $m \leftarrow \text{RandomMask}(r)$  ▷ Generate spatial mask
4: $q_u^* \leftarrow \text{Teacher}(x_u)$  ▷ Clean teacher prediction
5: $p_u \leftarrow \text{Student}(x_u, m)$  ▷ Masked student encoding
6: $p_v \leftarrow \text{Student}(x_v)$  ▷ Augmented student prediction
7: $\mathcal{L} \leftarrow \lambda_{\text{mim}}\mathcal{L}_{\text{mim}}(p_u, q_u^*, m) + \mathcal{L}_{\text{cons}}(p_v, q_u^*)$
8: $\theta \leftarrow \theta - \eta\nabla_\theta\mathcal{L}$  ▷ Single gradient step
9: $\theta^* \leftarrow \text{DualRateEMA}(\theta^*, \theta)$  ▷ Update teacher

---

Algorithm 1 summarizes our complete update procedure. The method requires exactly three forward passes (one teacher, two students) and one backward pass per batch. This computational efficiency, combined with the synergy between global semantic alignment, local spatial adaptation, and layer-wise EMA control, creates a minimal yet highly effective CTTA approach. The contribution of each component is quantified in our ablation studies (Table 3).

## 4 EXPERIMENTS

### 4.1 EXPERIMENTAL SETUP

We follow the continual test-time adaptation protocol established by Wang et al. (2022), where a source-trained network adapts online to a stream of fifteen corruption types at maximum severity without access to source data or ground-truth labels. Our evaluation spans ImageNet-C, CIFAR-10-C, and CIFAR-100-C benchmarks Hendrycks & Dietterich (2019), with experiments initialized from ViT-B/16 weights pre-trained on the corresponding source domains, following Liu et al. (2024a;b).

For consistency with recent work Liu et al. (2024a;b), we resize images to $224\times224$ for ImageNet-C and $384\times384$ for both CIFAR datasets. All experiments utilize fixed hyperparameters $\lambda_{\text{mim}} = 6$ and a masking ratio of $r = 0.5$, employing identical augmentation strategies for generating the student view $x_v$.

**Dataset-specific tuning.** We optimize using SGD with our dual-rate EMA where $(\alpha_{\text{attn}}, \alpha_{\text{weight}})$ are $(0.9, 0.99)$ for ImageNet-C and CIFAR-10-C, and $(0.999, 0.9999)$ for CIFAR-100-C. Learning rates are $\eta = 1\times10^{-5}$ for ImageNet-C and $\eta = 5\times10^{-5}$ for CIFAR datasets.

**Universal configuration (MiDEA-U).** To demonstrate robustness, we also evaluate a single hyper-parameter configuration across all datasets: $\eta = 1\times10^{-5}$, $\alpha_{\text{attn}} = 0.9$, and $\alpha_{\text{weight}} = 0.99$, while maintaining the same $\lambda_{\text{mim}} = 6$ and $r = 0.5$.

### 4.2 MAIN RESULTS

**Performance across corruption benchmarks.** MiDEA achieves substantial improvements over both non-adaptive baselines and state-of-the-art CTTA methods across all evaluated datasets. On ImageNet-C, MiDEA attains 38.1% error rate, surpassing the frozen source model by 18 percentage points and outperforming the previous best method (Continual-MAE) by 5 percentage points (Table 1). Notably, MiDEA achieves the lowest error on 9 out of 15 corruption types, compared to only 4 for the runner-up method.

Table 1: ImageNet-C tuned version, error (%) — lower is better. Columns show individual corruption types from ImageNet-C benchmark. Best results are indicated in **bold**.

| Method | REF | Gaussian | shot | impulse | defocus | glass | motion | zoom | snow | frost | fog | brightness | contrast | elastic trans. | pixelate | jpeg | Avg |
|---|---|---|---|---|---|---|---|---|---|---|---|---|---|---|---|---|---|
| Source | ICLR2021 | 53.0 | 51.8 | 52.1 | 68.5 | 78.8 | 58.5 | 63.3 | 49.9 | 54.2 | 57.7 | 26.4 | 91.4 | 57.5 | 38.0 | 36.2 | 55.8 |
| Pseudo-label | ICML2013 | 45.2 | 40.4 | 41.6 | 51.3 | 53.9 | 45.6 | 47.7 | 40.4 | 45.7 | 93.8 | 98.5 | 99.9 | 99.9 | 98.9 | 99.6 | 66.8 |
| TENT-continual | ICLR2021 | 52.2 | 48.9 | 49.2 | 65.8 | 73.0 | 54.5 | 58.4 | 44.0 | 47.7 | 50.3 | 23.9 | 72.8 | 55.7 | 34.4 | 33.9 | 51.0 |
| CoTTA | CVPR2022 | 52.9 | 51.6 | 51.4 | 68.3 | 78.1 | 57.1 | 62.0 | 48.2 | 52.7 | 55.3 | 25.9 | 90.0 | 56.4 | 36.4 | 35.2 | 54.8 |
| VDP | AAAI2023 | 52.7 | 51.6 | 50.1 | 58.1 | 70.2 | 56.1 | 58.1 | 42.1 | 46.1 | 45.8 | 23.6 | 70.4 | 54.9 | 34.5 | 36.1 | 50.0 |
| VIDA | CVPR2024 | 47.7 | 42.5 | 42.9 | 52.2 | 56.9 | 45.5 | 48.9 | 38.9 | 42.7 | 40.7 | 24.3 | 52.8 | 49.1 | 33.5 | 33.1 | 43.4 |
| Cont-MAE | AAAI2023 | 46.3 | 41.9 | 42.5 | 51.4 | 54.9 | 43.3 | 40.7 | **34.2** | **35.8** | 64.3 | **23.4** | 60.3 | 37.5 | **29.2** | **31.4** | 42.5 |
| **MiDEA** | **Proposed** | 45.1 | 39.2 | 40.6 | 46.8 | 46.2 | 40.5 | 39.2 | 34.6 | 36.0 | 35.6 | 25.0 | 42.1 | 37.7 | 30.4 | 32.2 | **38.1** |

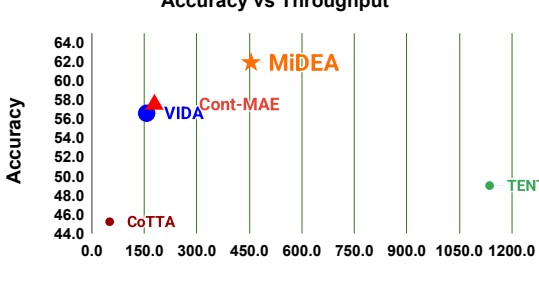

Figure 3: **Accuracy–throughput trade-off on ImageNet-C** (ViT-B/16, batch 8, single RTX 3080). MiDEA achieves 62% accuracy at 450 images/s, 3× faster and +5pp more accurate than top baselines (Continual-MAE, VIDA).

Similar trends emerge on CIFAR benchmarks. On CIFAR-10-C, MiDEA demonstrates a 17.6 percentage point improvement over the baseline and maintains a 2.5 percentage point margin over Continual-MAE, achieving best performance on 14 out of 15 corruptions. On the more challenging CIFAR-100-C dataset with its larger label space, MiDEA still delivers an 11.4 percentage point gain over the baseline and surpasses Continual-MAE by 2 percentage points, ranking first on 9 corruptions and second on only 3 (see Appendix D, for detailed per-corruption results). Results on ImageNet-C and CIFAR-100-C are averaged over 5 random seeds (std. ¡0.2); see Appendix C.

Table 2: Universal (**uni.**) vs. per–dataset (**tuned**) hyper–parameters. Lower error,↓; higher Avg–Gain,↑. While Versatile achieves 8.9% error on CIFAR-10-C, our method remains competitive across CIFAR-10/100 and surpasses it on ImageNet-C.

| Method | REF | Config | IN↓ | C10↓ | C100↓ | Avg↓ | Gain↑ |
|---|---|---|---|---|---|---|---|
| Source | ICLR2021 | – | 55.8 | 28.1 | 35.4 | 39.8 | 0.0 |
| TENT-continual | ICLR2021 | tuned | 51.0 | 23.5 | 32.1 | 35.5 | 4.3 |
| CoTTA | CVPR2022 | tuned | 54.8 | 24.6 | 34.8 | 38.0 | 1.8 |
| VDP | AAAI2023 | uni. | 50.0 | 24.1 | 32.0 | 35.4 | 4.4 |
| VIDA | ICLR2024 | tuned | 43.4 | 20.7 | 27.3 | 30.5 | 9.3 |
| Continual-MAE | CVPR2024 | tuned | 42.5 | 12.6 | 26.4 | 27.2 | 12.6 |
| Versatile | CVPR2024 | tuned | 42.7 | **8.9** | 24.0 | 25.2 | 14.6 |
| **MiDEA-U** | **Proposed** | uni. | 40.2 | 11.5 | **27.0** | 26.2 | 13.8 |
| **MiDEA** | **Proposed** | tuned | 38.1 | 10.3 | 24.7 | 24.4 | 15.7 |

**Universal hyperparameter robustness.** The universal configuration (MiDEA-U) demonstrates remarkable generalization, achieving error rates of 40.2%, 11.5%, and 27.0% on ImageNet-C, CIFAR-10-C, and CIFAR-100-C, respectively (Table 2). This single hyperparameter set not only maintains state-of-the-art performance but also exceeds the best individually tuned baselines by 1 percentage point on average. Across all datasets, MiDEA-U achieves a mean improvement of 13.8 percentage points compared to 12.6 for Continual-MAE and 9.3 for VIDA.

Versatile Yang et al. (2024a) reports strong CTTA results of 42.7% on ImageNet-C, 8.9% on CIFAR-10-C, and 24.0% on CIFAR-100-C. Relative to these numbers, MiDEA shows a clear advantage on ImageNet-C (38.1% vs. 42.7%). On CIFAR-100-C the methods are effectively tied: MiDEA reports 24.7% vs. 24.0%, and our source baseline is 0.6 points weaker, making the relative gains nearly identical. Versatile holds a small edge on CIFAR-10-C (8.9% vs. 10.3%) by 1.4 points.

Methodologically, Versatile derives its reliability and diversity signals from batch-level means and calibrates low-confidence samples using source-feature KNN within the current stage. In contrast,

MiDEA is decoder-free and single-teacher-pass, employs masked local self-distillation with ViT patch embeddings, and remains effective at batch size 1, yielding higher throughput in memory- and latency-limited settings.

Table 3: Ingredient analysis with the *universal* configuration ($\eta = 10^{-5}, \alpha_\mathrm{w} = 0.99, \alpha_\mathrm{attn} = 0.9$). IN = ImageNet-C, C10 = CIFAR-10-C, C100 = CIFAR-100-C. ✓ = component present.

| 2V | EMA$_2$ | MIM | IN↓ | C10↓ | C100↓ | Avg↓ | Gain↑ |
|----|---------|-----|------|------|-------|------|-------|
| ✓ | – | – | 44.1 | 17.9 | 29.2 | 30.4 | 9.6 |
| ✓ | ✓ | – | 40.9 | 13.3 | 30.1 | 28.1 | 11.9 |
| ✓ | ✓ | ✓ | 40.2 | 11.5 | 27.0 | 26.2 | 13.8 |

**Computational efficiency.** Beyond accuracy gains, MiDEA offers significant computational advantages. Processing 454 images per second at batch size 8, MiDEA runs three times faster than CoTTA, Continual-MAE, and VIDA without sacrificing accuracy (Figure 3). The advantage increases to 6x when VIDA and Continual-MAE use stochastic restoration (see Appendix B for details). Even under strict latency constraints with batch size 1, MiDEA maintains effectiveness with only 0.1 percentage point degradation (38.2% error), confirming the robustness of our batch-normalization-free update mechanism.

## 4.3 ABLATION ANALYSIS

We systematically validate each component of MiDEA using the universal hyperparameter configuration to ensure fair comparison across design choices.

**Component contribution analysis.** Table 3 reveals that each of MiDEA's three components provides distinct, complementary benefits. The global two-view consistency loss alone (2V) reduces error by 9.6 percentage points compared to the frozen source model, already surpassing VIDA's overall gain across datasets. This confirms that our teacher-student framework with augmented views provides a strong semantic adaptation signal.

Adding the dual-rate EMA mechanism (2V + EMA$_2$) contributes an additional 2.3 percentage points improvement by enabling attention layers to adapt quickly while stabilizing convolutional and MLP weights against catastrophic forgetting. Crucially, this benefit comes at zero computational overhead.

Finally, incorporating the local masked-token distillation loss (2V + EMA$_2$ + MIM) yields an additional 1.9 percentage point gain, completing the full MiDEA performance with a total improvement of 13.8 percentage points.

Note that our 2V baseline is conceptually related to the two-view loss in RMT Doebler et al. (2023). However, MiDEA does not attempt to reproduce the full RMT pipeline, which couples replay buffers, ensembling, and contrastive losses. Our work instead focuses on ViT-based CTTA, where we design and tune a holistic recipe, comprising augmentation placement, loss weighting, and dual-rate EMA scheduling, validated consistently across three datasets. This makes the approach distinct from prior work despite sharing the high-level idea of two-view consistency.

**Dual-rate EMA necessity.** To verify whether a single well-tuned EMA rate could match our dual-rate approach, we conduct a comprehensive sweep over single decay values (Table 4b). Results show that no single rate within a matched budget matches the dual-rate performance, with the best single configuration still lagging by an average of 1.8 points. Rates that are too conservative or too aggressive both lead to degradation and forgetting, supporting our hypothesis that heterogeneous EMA time scales are beneficial.

**Design choices validation.** Additional ablations confirm the robustness of our design decisions. Our choice of 50% random masking proves near-optimal on ImageNet-C (Table 4a), outperforming both sparse (30%) and heavy (70%) masking strategies by 2–3 percentage points. Likewise, our loss weighting scheme, $\lambda_\mathrm{mim} = 6$ for local masked-patch alignment and for global consistency, yields the best performance (Table 4c). The batch size robustness experiment demonstrates that our approach generalizes beyond the training configuration, maintaining effectiveness even under strict inference constraints.

Table 4: Ablation study on MiDEA hyperparameters on ImageNet-C (severity 5). (a) Mask ratio, (b) EMA schedule (dual vs. single rate), and (c) MIM loss weight. IN = ImageNet-C, C10 = CIFAR-10-C, C100 = CIFAR-100-C.

(a) Mask ratio $r$

| $r$ | mCE↓ |
|------|------|
| 0.30 | 40.9 |
| **0.50** | **38.1** |
| 0.70 | 39.2 |

(b) EMA scheme

| Scheme | IN↓ | C10↓ | C100↓ | Avg↓ |
|--------|-----|------|-------|------|
| Single 0.999 | 44.5 | 16.3 | 28.9 | 29.9 |
| Single 0.997 | 43.4 | 14.0 | 33.0 | 30.1 |
| Single 0.995 | 42.8 | 13.2 | 35.8 | 30.6 |
| Single 0.993 | 42.9 | 12.6 | 38.2 | 31.2 |
| Single 0.990 | 42.4 | 11.8 | 43.8 | 32.7 |
| Dual (0.99,0.999) | **40.9** | **13.3** | **30.1** | **28.1** |

(c) MIM weight $\lambda_{\mathrm{mim}}$

| $\lambda_{\mathrm{mim}}$ | mCE↓ |
|------|------|
| 0 | 38.9 |
| 3 | 40.7 |
| **6** | **38.1** |
| 9 | 40.2 |

## 5 DISCUSSION

MiDEA demonstrates that a minimal teacher–student recipe, carefully tuned for ViTs, can achieve state-of-the-art CTTA performance under strict efficiency constraints. Here, we reflect on its design choices, its relation to prior work, and its limitations.

**Dual-rate EMA.** CTTA demands both rapid adaptation and stability. Our dual-rate EMA updates attention layers faster while keeping MLP weights conservative; ablations show no single global rate recovers this balance.

**Global and local self-distillation.** Global two-view consistency complements masked local alignment: masking prevents trivial patch matching, forces context inference, and reduces overfitting to noisy pseudo-labels.

**Relation to Robust Mean Teacher (RMT).** Our two-view baseline is conceptually related to RMT Doebler et al. (2023), which also employs teacher–student consistency. However, RMT couples this loss with replay buffers, contrastive objectives, and teacher–student ensembling on CNNs. MiDEA instead focuses on ViT-based CTTA with a decoder-free design, and our gains stem from combining the two-view baseline with dual-rate EMA and masked local alignment. This positions MiDEA as a distinct, lightweight recipe rather than a reproduction of RMT.

**Efficiency and deployment.** With only one teacher and two student passes per batch, MiDEA runs 3x faster than multi-pass methods such as Continual-MAE and VIDA, while uniquely retaining accuracy down to batch size 1. This makes MiDEA practical for real-time, resource-constrained use.

**Limitations and future work.** Our design remains empirically motivated; a formal theory for why masked local alignment and dual-rate EMA reduce forgetting is still lacking. MiDEA is also currently evaluated on ViTs for classification. Extending the framework to dense prediction and exploring dynamic EMA schedules are promising next steps.

## 6 CONCLUSION

We introduce **MiDEA: Masked-Image Modeling with Dual-EMA Adaptation**, an efficient framework for continual test-time adaptation (CTTA) addressing catastrophic forgetting and error accumulation. MiDEA's innovation lies in three synergistic components: **global two-view consistency, local masked-token distillation** (decoder-free patch alignment), and **dual-rate EMA** balancing plasticity and stability. This design ensures rapid adaptation while preserving knowledge. Empirically, MiDEA achieves 38.1% error on ImageNet-C, 18 percentage points better than frozen models and 5 points above prior SOTA, running 3x faster and stable at batch size 1, proving its practical value for real-time deployment. Overall, MiDEA provides a lightweight, reproducible recipe for ViT-based CTTA, setting a new benchmark for efficiency-oriented robustness.

## ETHICS STATEMENT

This work targets maintaining AI model accuracy under real-world distribution shifts. Such robustness could enable expanded surveillance capabilities. Deployments should follow privacy laws and include independent review.

The main technical risk is over-reliance on adaptation when distribution shifts fall outside the evaluated corruptions (weather, noise, blur), which can cause accuracy drift. We recommend guardrails including confidence monitoring and safe fallback to the fixed source model, especially in safety-critical settings.

Our experiments use Vision Transformers. Other architectures should be validated before deployment. This study involved no human subjects or sensitive personal data, and we anticipate no additional third-party risks beyond those noted above.

## REPRODUCIBILITY STATEMENT

We provide code, configuration files, and runnable scripts to reproduce all results. The root `README.md` details environment setup, dataset preparation for ImageNet-C/CIFAR-C, and exact command lines. All hyperparameters match those reported in the paper.

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

## A  ADDITIONAL RESULTS

This supplementary material provides comprehensive experimental details and additional analyses that complement the main results. We present four key contributions: (1) efficiency analysis clarifying conditions under which MiDEA achieves up to 6× throughput advantage over competing methods, (2) statistical robustness validation across multiple random seeds, (3) detailed per-corruption performance breakdowns for all evaluated benchmarks, and (4) complete implementation details including augmentation specifications and code reproduction instructions.

## B  EFFICIENCY AND BATCH-SIZE ANALYSIS

### B.1  SPEED–ACCURACY TRADE-OFF (BATCH 8)

As established in Section 4.2 (Main Results), MiDEA achieves a 3× throughput gain over prior multi-pass CTTA methods at batch size 8. A closer look reveals that the actual speed advantage is often larger in practice. This is due to the *stochastic restore (SR)* mechanism used by several baselines, such as Continual-MAE Liu et al. (2024a) (line 157 of `continual_mae.py`) [1] and VIDA Liu et al. (2024b) (line 142 of `cifar/vida.py`) [2], which is not reflected in their original papers but is implemented in their released code. When SR is enabled, these methods experience substantial slowdowns: for example, VIDA drops from 156 to 93 images/s, and Continual-MAE from 179 to 79 images/s (Table 5).

**MiDEA does not rely on SR** or any auxiliary replay mechanism and thus avoids this bottleneck entirely. As a result, it consistently achieves up to **6×** **higher throughput** than these baselines, reaching **455 images/s** at batch size 8 with just **4.9 GB** of GPU memory on a single RTX 3080, while maintaining superior accuracy.

Table 5: Efficiency analysis on ImageNet-C (ViT-B/16, RTX 3080): MiDEA achieves 6× higher throughput and uses less than 5GB memory while maintaining accuracy even at batch size 1. See text for details.

| Method | Batch | Memory (GB) | FPS | Accuracy (%) | Configuration |
|--------|-------|-------------|-----|--------------|---------------|
| TENT | 8 | 3.0 | 1136 | 49.0 | Standard |
| CoTTA | 8 | 5.7 | 50 | 45.2 | Standard |
| VIDA (SR disabled) | 8 | 4.6 | 156 | 56.6 | Single extra pass |
| VIDA (SR enabled) | 8 | 4.6 | 93 | 56.6 | With restoration |
| Cont-MAE (SR disabled) | 8 | 4.0 | 179 | 57.5 | Single extra pass |
| Cont-MAE (SR enabled) | 8 | 4.0 | 79 | 57.5 | With restoration |
| **MiDEA** | 8 | **4.9** | **455** | **61.9** | Dual-rate EMA |
| **MiDEA** | 1 | **2.7** | **185** | **62.1** | Low-latency mode |

---

[1] `https://github.com/RanXu2000/continual-mae/blob/ea25f1c525131aeba253d6f34a4b976ed06c8ffe/cifar/continual_mae.py#L157`

[2] `https://github.com/Yangsenqiao/vida/blob/c26b0db2eac6886856aec64eb2032f68654d49f5/cifar/vida.py#L142`

Table 6: Batch size robustness on ImageNet-C: MiDEA error rates (%). Accuracy remains stable down to batch size 1, confirming suitability for low-latency deployment.

| LR | Batch | Gau | Shot | Imp | Def | Glass | Mot | Zoom | Snow | Frost | Fog | Brt | Cnt | Elas | Pix | JPEG | **Mean** |
|----|-------|-----|------|-----|-----|-------|-----|------|------|-------|-----|-----|-----|------|-----|------|----------|
| $1 \times 10^{-6}$ | 1 | 45.42 | 39.66 | 40.18 | 47.04 | 46.16 | 39.80 | 39.38 | 35.20 | 35.62 | 33.92 | 23.66 | 42.86 | 38.64 | 29.24 | 31.28 | 37.9 |
| $3 \times 10^{-6}$ | 2 | 45.68 | 38.90 | 39.54 | 47.46 | 47.20 | 39.86 | 39.54 | 35.18 | 35.60 | 36.06 | 24.06 | 42.48 | 39.90 | 30.10 | 31.34 | 38.2 |
| $6 \times 10^{-6}$ | 4 | 46.60 | 38.76 | 39.24 | 48.58 | 48.56 | 40.46 | 42.40 | 36.12 | 36.06 | 36.78 | 24.12 | 42.54 | 42.62 | 30.90 | 31.80 | 39.0 |
| $1 \times 10^{-5}$ | 8 | 45.10 | 39.24 | 40.60 | 46.76 | 46.18 | 40.46 | 39.22 | 34.62 | 36.04 | 35.60 | 24.98 | 42.10 | 37.74 | 30.40 | 32.22 | 38.1 |

## B.2 STOCHASTIC RESTORE

Stochastic Restore (SR) was introduced by CoTTA Wang et al. (2022) as a mechanism to prevent catastrophic forgetting and error accumulation during continual adaptation. SR periodically resets a random subset of BatchNorm statistics and selected model weights to their pre-trained values for every test sample. While this helps maintain stability, it introduces computational overhead by requiring duplicate parameter storage and conditional copying operations during inference.

## B.3 BATCH-SIZE SENSITIVITY

To further assess MiDEA's efficiency in latency-constrained scenarios, we analyze its performance at varying batch sizes. As shown in Table 5, MiDEA maintains high throughput even at batch size 1, processing **185 images/s** using only **2.7 GB** of GPU memory—making it suitable for real-time or low-latency deployment settings. Notably, accuracy remains stable in this regime, with a slight increase from 61.9% to 62.1%, highlighting MiDEA's robustness to batch-size changes.

Additional sweeps over batch sizes (Table 6) confirm that MiDEA's performance scales gracefully, with consistent accuracy and predictable computational cost across a wide range of settings. This distinguishes MiDEA from existing methods that are tightly coupled to larger batch sizes or incur significant accuracy drops when operating under limited compute.

# C STATISTICAL ROBUSTNESS ACROSS MULTIPLE SEEDS

We validate MiDEA's performance reliability through five independent random seeds to ensure reported gains are not due to fortuitous initialization. Tables 7 and 8 present comprehensive results for ImageNet-C and CIFAR-100-C respectively.

MiDEA demonstrates exceptional stability with minimal variance across seeds: ImageNet-C shows mean error variation of only 0.1 percentage points, while CIFAR-100-C exhibits similarly low variability. Per-corruption fluctuations remain well within 0.3 percentage points, confirming consistent performance regardless of random initialization.

This statistical robustness validates the reliability of our reported improvements and demonstrates MiDEA's practical deployability.

**Experimental Configuration.** ImageNet-C experiments use batch size 8, learning rate 1e-5, and dual-rate EMA parameters $(\alpha_{\text{attn}}, \alpha_{\text{weight}}) = (0.9, 0.99)$. CIFAR-100-C experiments use batch size 4, learning rate 5e-5, and dual-rate EMA parameters $(\alpha_{\text{attn}}, \alpha_{\text{weight}}) = (0.999, 0.9999)$.

Table 7: **Statistical robustness analysis:** MiDEA error rates (%) on **ImageNet-C** over five independent random seeds. The table highlights the method's reproducibility, reporting per-seed results across all 15 corruption types. The "Avg." column summarizes the mean error for each seed. Lower error values reflect stronger robustness. **The consistently low standard deviation ($\leq 0.1$ pp)) across seeds further supports the reliability and stability of MiDEA's improvements.**

| Seed | Gaussian | Shot | Impulse | Defocus | Glass | Motion | Zoom | Snow | Frost | Fog | Bright. | Contr. | Elastic | Pixelate | JPEG | Avg. |
|------|----------|------|---------|---------|-------|--------|------|------|-------|-----|---------|--------|---------|----------|------|------|
| 1 | 45.66 | 40.00 | 41.50 | 46.52 | 46.28 | 40.56 | 39.54 | 35.46 | 35.96 | 35.48 | 24.84 | 43.94 | 38.68 | 30.42 | 32.02 | 38.5 |
| 2 | 45.60 | 39.62 | 40.56 | 46.20 | 46.38 | 40.74 | 39.44 | 34.64 | 35.78 | 35.62 | 25.02 | 44.60 | 39.52 | 30.98 | 32.42 | 38.5 |
| 3 | 46.00 | 40.24 | 40.86 | 46.64 | 46.24 | 40.44 | 39.50 | 35.04 | 35.96 | 35.50 | 24.86 | 44.90 | 39.10 | 30.48 | 31.94 | 38.5 |
| 4 | 46.30 | 40.10 | 40.80 | 46.76 | 46.42 | 41.08 | 39.32 | 35.04 | 36.12 | 35.32 | 24.82 | 44.84 | 39.18 | 30.66 | 32.62 | 38.6 |
| 5 | 45.78 | 40.12 | 41.30 | 46.52 | 45.46 | 40.44 | 38.92 | 34.98 | 35.52 | 35.70 | 24.62 | 44.30 | 38.36 | 30.42 | 32.32 | 38.3 |
| **Mean** | 45.87 | 40.02 | 41.00 | 46.53 | 46.16 | 40.65 | 39.34 | 35.03 | 35.87 | 35.52 | 24.83 | 44.52 | 38.97 | 30.59 | 32.26 | 38.48 |
| **Std.** | 0.26 | 0.21 | 0.34 | 0.19 | 0.35 | 0.24 | 0.22 | 0.26 | 0.20 | 0.13 | 0.13 | 0.36 | 0.40 | 0.21 | 0.25 | 0.10 |

Table 8: **Statistical stability analysis**: MiDEA error rates (%) on **CIFAR-100-C** across five independent random seeds. This table demonstrates the reproducibility and robustness of our method by showing consistent performance across different initializations. Each row represents results from one random seed, with columns showing error rates for all 15 corruption types. The "Avg." column reports the mean error across all corruption types for each seed. Lower values indicate better performance. **The minimal variance across seeds (standard deviation ($\leq 0.2$ pp)) confirms the statistical reliability of our reported improvements**.

| Seed | Gaussian | Shot | Impulse | Defocus | Glass | Motion | Zoom | Snow | Frost | Fog | Bright. | Contr. | Elastic | Pixelate | JPEG | Avg. |
|------|----------|------|---------|---------|-------|--------|------|------|-------|-----|---------|--------|---------|----------|------|------|
| 1 | 47.62 | 38.56 | 22.70 | 24.08 | 36.26 | 22.58 | 18.82 | 18.58 | 17.34 | 20.66 | 14.48 | 15.76 | 25.38 | 23.80 | 24.74 | 24.80 |
| 2 | 48.18 | 38.94 | 22.60 | 23.24 | 35.58 | 22.26 | 18.50 | 17.78 | 16.54 | 20.12 | 13.36 | 15.02 | 24.02 | 22.92 | 24.70 | 24.30 |
| 3 | 47.56 | 39.30 | 23.34 | 23.78 | 35.60 | 22.38 | 18.56 | 18.52 | 17.48 | 20.58 | 14.18 | 15.98 | 25.28 | 23.30 | 24.66 | 24.70 |
| 4 | 47.74 | 39.48 | 23.22 | 23.96 | 35.28 | 22.56 | 18.76 | 18.30 | 17.28 | 21.38 | 14.10 | 15.92 | 24.74 | 24.70 | 25.02 | 24.80 |
| 5 | 46.82 | 37.78 | 23.18 | 23.48 | 36.94 | 22.20 | 18.66 | 18.66 | 17.98 | 20.48 | 14.30 | 15.56 | 25.30 | 23.12 | 24.22 | 24.60 |
| **Mean** | 47.58 | 38.81 | 23.01 | 23.71 | 35.93 | 22.40 | 18.66 | 18.37 | 17.32 | 20.64 | 14.08 | 15.65 | 24.94 | 23.57 | 24.67 | 24.64 |
| **Std.** | 0.44 | 0.60 | 0.30 | 0.31 | 0.60 | 0.15 | 0.12 | 0.32 | 0.46 | 0.41 | 0.38 | 0.35 | 0.51 | 0.64 | 0.26 | 0.19 |

# D FULL PER-CORRUPTION RESULTS

We provide detailed per-corruption breakdowns for both dataset-specific and universal (hyperparameter-agnostic) configurations discussed in Section 4.2 (Main Results).

The first presents detailed results for CIFAR-10-C using dataset-specific hyperparameter tuning, where parameters are optimized individually for each dataset following the approach used by prior methods such as VIDA and Continual-MAE. The second demonstrates the robustness of our universal configuration (MiDEA-U), which employs a single, fixed set of hyperparameters applied uniformly across all three datasets (ImageNet-C, CIFAR-10-C, and CIFAR-100-C). This universal approach contrasts with existing methods that require dataset-specific parameter tuning, highlighting MiDEA's ability to achieve strong performance without per-dataset optimization.

Table 9: Detailed per-corruption breakdown: CIFAR-10-C error rates (%) across all 15 corruption types at severity level 5. This table provides comprehensive results for individual corruption categories. Each column represents a specific corruption type, allowing for detailed analysis of MiDEA's performance across different types of image degradation. Lower error rates indicate better performance.

| Method | REF | Gau | Shot | Imp | Def | Glass | Mot | Zoom | Snow | Frost | Fog | Brt | Cnt | Elas | Pix | JPEG | Avg | Gain |
|--------|-----|-----|------|-----|-----|-------|-----|------|------|-------|-----|-----|-----|------|-----|------|-----|------|
| Source | ICLR2021 | 60.1 | 53.2 | 38.3 | 19.9 | 35.5 | 22.6 | 18.6 | 12.1 | 12.7 | 22.8 | 5.3 | 49.7 | 23.6 | 24.7 | 23.1 | 28.1 | 0.0 |
| Our Source | | 59.4 | 52.3 | 37.8 | 19.3 | 35.4 | 22.3 | 18.3 | 12.3 | 13.0 | 23.0 | 5.5 | 49.4 | 23.1 | 24.0 | 23.2 | 27.9 | 0.3 |
| Pseudo-label | ICML2013 | 59.8 | 52.5 | 37.2 | 19.8 | 35.2 | 21.8 | 17.6 | 11.6 | 12.3 | 20.7 | 5.0 | 41.7 | 21.5 | 25.2 | 22.1 | 26.9 | 1.2 |
| TENT-continual | ICLR2021 | 57.7 | 56.3 | 29.4 | 16.2 | 35.3 | 16.2 | 12.4 | 11.0 | 11.6 | 14.9 | 4.7 | 22.5 | 15.9 | 29.1 | 19.5 | 23.5 | 4.6 |
| CoTTA | CVPR2022 | 58.7 | 51.3 | 33.0 | 20.1 | 34.8 | 20.0 | 15.2 | 11.1 | 11.3 | 18.5 | 4.0 | 34.7 | 18.8 | 19.0 | 17.9 | 24.6 | 3.6 |
| VDP | AAAI2023 | 57.5 | 49.5 | 31.7 | 21.3 | 35.1 | 19.6 | 15.1 | 10.8 | 10.3 | 18.1 | 4.0 | 27.5 | 18.4 | 22.5 | 19.9 | 24.1 | 4.1 |
| VIDA | CVPR2024 | 52.9 | 47.9 | 19.4 | 11.4 | 31.3 | 13.3 | 7.6 | 7.6 | 9.9 | 12.5 | 3.8 | 26.3 | 14.4 | 33.9 | 18.2 | 20.7 | 7.5 |
| Cont-MAE | AAAI2023 | **30.6** | 18.9 | 11.5 | 10.4 | 22.5 | 13.9 | 9.8 | 6.6 | 6.5 | 8.8 | 4.0 | 8.5 | 12.7 | 9.2 | 14.4 | 12.6 | 15.6 |
| **MiDEA** | **Proposed** | 42.7 | **15.4** | **8.7** | **8.5** | **12.7** | **8.0** | **6.2** | **6.2** | **5.3** | **6.2** | **3.8** | **4.7** | **9.2** | **7.1** | **9.8** | **10.3** | **17.6** |

Table 10: Universal configuration robustness: MiDEA-U detailed per-corruption results across all evaluated benchmarks using identical hyperparameters. This table demonstrates MiDEA's ability to achieve strong performance across diverse datasets without dataset-specific tuning, highlighting the robustness of our universal configuration approach. Results are shown for all 15 corruption types on ImageNet-C, CIFAR-10-C and CIFAR-100-C (error rates, lower is better)). The consistent strong performance across different image resolutions (224×224 for ImageNet, 32×32 for CIFAR), number of classes (1000, 10, 100), and data characteristics using the same hyperparameter set validates MiDEA's generalizability and practical deployment advantages. This universal applicability eliminates the need for costly hyperparameter search when adapting to new domains or datasets.

| Dataset | Gau | Shot | Imp | Def | Glass | Mot | Zoom | Snow | Frost | Fog | Brt | Cnt | Elas | Pix | JPEG | **Mean** |
|---------|-----|------|-----|-----|-------|-----|------|------|-------|-----|-----|-----|------|-----|------|----------|
| ImageNet-C | 46.88 | 39.88 | 39.90 | 51.96 | 50.26 | 42.04 | 43.34 | 36.48 | 36.76 | 37.70 | 27.14 | 45.50 | 40.90 | 31.74 | 32.70 | 40.20 |
| CIFAR-10-C | 48.12 | 27.78 | 11.32 | 7.90 | 15.60 | 7.38 | 5.00 | 6.18 | 4.72 | 5.94 | 3.04 | 3.54 | 8.72 | 8.48 | 8.38 | 11.50 |
| CIFAR-100-C | 39.82 | 31.96 | 25.20 | 26.74 | 35.18 | 26.66 | 22.94 | 23.58 | 22.88 | 24.38 | 18.82 | 20.34 | 29.10 | 26.68 | 30.84 | 27.00 |

**Baseline Methodology.** To ensure fair comparison under identical experimental conditions, we reproduce baseline results using the same model architecture and pre-trained weights as the original methods. While these reproduced baselines should theoretically match the originally reported values, we observe minor differences (less than 0.6,pp) likely due to implementation details, hardware variations, or software version differences. Therefore, all gain calculations are computed relative to our reproduced "Source" baselines rather than originally reported numbers, ensuring coherent and transparent comparisons under our exact experimental setup.

**Experimental Configuration.** CIFAR-10-C images are resized to 384×384 following recent work Liu et al. (2024a;b). MiDEA employs dataset-specific tuning with SGD optimizer, dual-rate EMA parameters ($\alpha_{\text{attn}} = 0.9, \alpha_{\text{weight}} = 0.99$), and learning rate $\eta = 5 \times 10^{-5}$.

**Universal configuration (MiDEA-U).** Table 10 reports the complete per-corruption results for MiDEA evaluated with a single, universal set of hyperparameters applied uniformly across all datasets. In contrast to approaches that rely on dataset-specific tuning, MiDEA-U achieves strong generalization and robustness using fixed settings: a learning rate of $\eta = 3 \times 10^{-5}$, dual-rate EMA parameters ($\alpha_{attn} = 0.9, \alpha_{weight} = 0.99$), a local alignment loss weight of $\lambda_{mim} = 6$, and a masking ratio of $r = 0.5$. MiDEA-U therefore outperforms the best dataset-*tuned* baselines by **approximately 1 pp on average** while maintaining a **13.8 pp** margin over the frozen source.

# E IMPLEMENTATION AND AUGMENTATION DETAILS

We provide detailed specifications for MiDEA's data augmentation pipeline, which plays a crucial role in the global consistency objective. Our augmentation strategy balances computational efficiency with adaptation effectiveness through asymmetric processing: the teacher network processes clean input images while the student network receives strongly augmented views.

**Augmentation Pipeline.** The student network processes strongly augmented inputs to learn robust feature representations under challenging visual conditions, while the teacher network maintains stable reference predictions from clean, unaugmented images. This asymmetric processing strategy drives effective adaptation by encouraging the student to extract invariant semantic features despite visual perturbations.

**Global Consistency Impact.** The global consistency loss alone—using only this augmentation strategy without local masked modeling—achieves a substantial 9.6 percentage point error reduction over the frozen source baseline. This demonstrates the effectiveness of our augmentation-driven consistency objective as a core component of MiDEA's adaptation mechanism.

The complete augmentation specifications and ablation results are detailed in the following table 11.

Table 11: Student augmentation pipeline for global consistency. Teacher inputs are unaugmented.

| Augmentation | Parameters + note |
|---|---|
| RandomAffine | degrees $0°$
translate ($1/W \cdot 32$, $1/H \cdot 32$)
scale 0.98–1.02, bilinear, fill=0
*no rot.; $\leq 1/32$ shift; $\pm 2$ % zoom* |
| ColorJitterPro | brightness 0.6–1.4, contrast 0.7–1.3,
saturation 0.5–1.5, hue $\pm 0.06$,
gamma 0.7–1.3 |
| RandomGrayscale | $p = 0.2$ |

# F LLM USAGE DISCLOSURE

We used a large language model (ChatGPT) for minor grammar and phrasing edits only. All ideas, methods, experiments, and analyses are entirely our own.

