# OpenReview forum: "Fast and Stable Continual Test-Time Adaptation via Masked Modeling and Momentum-Guided Updates"
_ICLR.cc/2026/Conference — ICLR 2026 Conference Withdrawn Submission_

### Official Review · Reviewer_T5PU · 2025-10-30

**Soundness:** 1
**Presentation:** 1
**Contribution:** 1
**Rating:** 0
**Confidence:** 5

**Summary:**

This paper presents a continual test-time adaptation (CTTA) method designed to operate under limited computational resources. The proposed approach combines a teacher–student framework with masked image modeling. It requires one teacher and two student forward passes per batch. The method is evaluated on standard corruption benchmarks including ImageNet-C, CIFAR-10-C, and CIFAR-100-C.

**Strengths:**

1. The paper motivates the need for test-time adaptation well.
2. The authors present Figure 1 effectively, making it easy to understand.

**Weaknesses:**

1. There is still significant room for improvement in this paper.
   1. Figure 1: Could the four images be arranged horizontally? The caption is usually placed either above or below. In the current version, there is excessive blank space around the caption.
   2. Figure 2: The right margin is wider than the left. Could the figure be center-aligned?
   3. Figure 3: It would be better if the graph were redrawn using Python. The missing bottom horizontal axis line is unusual. The number of baselines is small relative to the figure size.
   4. Table 1: The label “proposed” seems awkward. Also, the font size of the numbers could be increased and the horizontal spacing between numbers reduced.
   5. Table 4: Could the empty space be filled? For examples of how to align multiple tables neatly, refer to the MAE [10] paper.
   6. Tables 1~4: The distance between each table and its caption is large.
   7. The spacing between paragraphs is wide throughout the paper, which leaves excessive blank areas.
   8. Several paragraphs end with only a few words on the final line, which looks unprofessional. Please adjust the text so that the last sentence of each paragraph extends to fill the line width (e.g., Lines 27, 72, 95, 201, 210, 234, 242, 395, 410, 431).
   9. The last page contains too much empty space. Additional ablation studies should be included.
2. A more novel approach needs to be developed. The paper does not provide new insights or findings from the proposed methodology.
   1. The teacher–student structure has been explored in [1~4], the variation of cross-entropy in [2], the use of masking in [5], and layer-wise techniques in [11].
   2. While the paper motivates the need for test-time adaptation, the proposed solution offers limited novelty and does not contribute new understanding.
3. The main goal of this paper is to reduce computational overhead. However, the proposed method still requires heavy operations such as dual inference of teacher and student networks. Compared with previous studies [6~9], it is difficult to consider the method efficient.



[1] Continual Test-Time Domain Adaptation CVPR 2022

[2] A Probabilistic Framework for Lifelong Test-Time Adaptation CVPR 2023

[3] Persistent Test-time Adaptation in Recurring Testing Scenarios NeurIPS 2024

[4] Improved Self-Training for Test-Time Adaptation CVPR 2024

[5] Test-Time Training with Masked Autoencoders NIPS 2022

[6] MECTA: Memory-Economic Continual Test-Time Model Adaptation ICLR 2023

[7] EcoTTA: Memory-Efficient Continual Test-time Adaptation via Self-distilled Regularization CVPR 2023

[8] BECoTTA: Input-dependent Online Blending of Experts for Continual Test-time Adaptation ICML 2024

[9] SURGEON: Memory-Adaptive Fully Test-Time Adaptation via Dynamic Activation Sparsity CVPR 2025

[10] Masked Autoencoders Are Scalable Vision Learners CVPR 2022

[11] Efficient Test-Time Model Adaptation without Forgetting ICLR 2022

**Questions:**

Could authors review the recommended papers above? Compared with their quality, the current paper has substantial room for improvement.

---

### Official Review · Reviewer_uuNd · 2025-10-30

**Soundness:** 2
**Presentation:** 1
**Contribution:** 2
**Rating:** 2
**Confidence:** 4

**Summary:**

This paper proposes MIDEA, a continual test-time adaptation (CTTA) framework for Vision Transformers, designed to be fast and stable. The method uses a teacher-student architecture and combines three components: 1) a global, two-view consistency loss between the teacher and student, 2) a decoder-free, local masked-token alignment loss to provide spatial regularization, and 3) a dual-rate Exponential Moving Average (EMA) update, which adapts attention layers faster than MLP layers to balance plasticity and stability. The authors claim this method achieves state-of-the-art performance and is 3x faster than competing methods.

**Strengths:**

The paper's objective of creating a computationally efficient and fast CTTA method is a practical and important goal for real-world deployment, especially the focus on maintaining performance at batch size 1. The core ideas, such as using a dual-rate EMA to treat attention and MLP layers differently, or employing decoder-free masked alignment for regularization, are conceptually interesting.

**Weaknesses:**

This paper appears to be in a very early draft stage and is not ready for publication. It suffers from significant issues in its experimental comparisons, internal consistency, and overall presentation.

1. Inconsistent Naming and Citations: The paper is riddled with inconsistencies that undermine its credibility. For example, the same method is referred to as "Cont-MAE" (Table 1), "Continual-MAE" (Table 2, Line 55), and "ADMA" (Line 160). Furthermore, the citation venues for key baselines conflict within the paper itself: Cont-MAE is listed as "AAAI2023" in Table 1 but "CVPR2024" in Table 2, while VIDA is listed as "CVPR2024" in Table 1 but "ICLR2024" in Table 2. This sloppiness makes the work feel unreliable.

2. Weak and Out-of-Date Baselines: The paper's core motivation is to provide a new state-of-the-art in fast CTTA, but this claim is based on a weak and selective set of baselines. More recent methods [1-4] need to be discussed. Especially if the main goal is an efficient CTTA method (Line 57), comparisons against more parameter-efficient, memory-efficient, and decoder-free methods are required [1-4], especially since teacher-student style CTTA is usually slow and inefficient[1,3]. This narrow focus on the teacher-student paradigm makes the contribution feel limited.

3. Selective Comparisons and Hyperparameter Sensitivity: The paper's claims are weakened by selective comparisons; for instance, the SOTA method Versatile is omitted from the main ImageNet-C table (Table 1). More concerningly, the paper presents "per-dataset" tuned hyperparameters. This is impractical for a true TTA setting, as hyperparameters must be determined before encountering any test data. This raises the question of how the "per-dataset" hyperparameters were determined. Moreover, the performance drop between the "per-dataset" and "universal" settings (e.g., a 2% drop in Table 2, where alphas change from 0.9999 to 0.99; also in Table 4b) raises the concern that the method is overly sensitive to its hyperparameters.

4. Poor Writing and Unclear Claims: The writing is often imprecise. For example, the paper's central efficiency claim is confusing; it implies a single or dual-pass method in the introduction ("single teacher pass"), but the method section (Line 196) clarifies it actually "requires two student and one teacher forward passes per batch (three total)." Additionally, Algorithm 1 is poorly formatted with no indentation and appears to be missing a for loop, making it difficult to understand the update process. The table and figure layouts are also unconventional (e.g., Table 4).


[1] BECoTTA: Input-dependent Online Blending of Experts for Continual Test-time Adaptation (ICML2024)

[2] Persistent Test-time Adaptation in Recurring Testing Scenarios (NeurIPS2024)

[3] DPCore: Dynamic Prompt Coreset for Continual Test-Time Adaptation (ICML 2025)

[4] Ranked Entropy Minimization for Continual Test-Time Adaptation  (ICML 2025)

**Questions:**

1. The 3-pass design (1 teacher, 2 student) is a key part of the method. How does its throughput (images/sec) compare to a simpler 3-pass baseline (e.g., running TENT 3 times) to isolate the actual overhead of the masked modeling and dual EMA updates?

2. In line 353, the notation in "std. 0.2" is confusing and possibly an error.

3. The local masked alignment (L_mim) is a key contribution. The ablation (Table 3) shows it adds an average improvement of ~1.9%. Can the authors provide more intuition or analysis on why matching only the masked tokens is superior to a simpler, decoder-free consistency loss across all patch tokens?

4. Providing a direct comparison of wall clock time will be helpful.

---

### Official Review · Reviewer_rSbz · 2025-10-31

**Soundness:** 3
**Presentation:** 3
**Contribution:** 2
**Rating:** 4
**Confidence:** 3

**Summary:**

MiDEA (Masked-image modeling with Dual-EMA Adaptation) tackles continual test-time adaptation (CTTA) for Vision Transformers using three components: (i) global two-view consistency between clean teacher and augmented student views, (ii) decoder-free local masked self-distillation matching patch embeddings on masked regions, and (iii) layer-wise dual-rate EMA where attention layers adapt faster ($\alpha_{\text{attn}} < \alpha_{\text{weight}}$) than MLPs. On ImageNet-C with ViT-Base, MiDEA achieves 38.1% error (18 percentage points below frozen baseline, 5 percentage points below Continual-MAE) at 454 images/s; 3× faster than multi-pass methods; while maintaining accuracy at batch size 1.

**Strengths:**

1. **Strong empirical results**: 5 percentage point improvement over Continual-MAE on ImageNet-C with comprehensive 3-benchmark evaluation and proper statistics.
2. **Systematic ablations**: Table 3 cleanly isolates each component; Table 4b shows no single EMA rate matches dual-rate.
3. **Practical efficiency**: 3× throughput (454 vs ~150 images/s) and batch-size-1 robustness (0.1 percentage point degradation) address real deployment constraints.
4. **Universal config (MiDEA-U)**: Fixed hyperparameters achieve 13.8 percentage point average gain across datasets, demonstrating robustness.

**Weaknesses:**

1. **No theoretical foundation**: Dual-rate EMA is purely empirical. Why should attention adapt at $\alpha=0.9$ vs. MLPs at $\alpha=0.99$? No gradient analysis, drift metrics, or forgetting rates provided.

2. **Insufficient novelty over RMT**: 2-view baseline is "conceptually related" to RMT yet no direct comparison. The 9.6 percentage point gain from 2-view alone (70% of total) suggests most improvement comes from RMT-like component, not novel contributions.

3. **Weak masking justification**: Cross-entropy loss on embeddings chosen arbitrarily. Only ratio ablated (Table 4a), not loss type. Modest 1.9 percentage point contribution suggests limited impact.

4. **No architectural generalization**: Claims about "ViT-based CTTA" require validation on ViT-S/L, Swin Transformers, or CNNs to test if dual-rate EMA is architecture-specific.

5. **Incomplete efficiency analysis**: Throughput reported but no FLOP counts or memory breakdown. How much speedup from single-teacher-pass vs. decoder removal vs. no batch-norm?

6. **Stability overclaimed**: "Reduces drift" stated but no drift metric ($\|\theta_t - \theta_0\|$, backward transfer) provided; only final accuracy.

7. **Gap with Versatile unexplored**: Why does Versatile beat MiDEA by 1.4 percentage points on CIFAR-10-C? Are reliability filtering and KNN calibration complementary?

**Questions:**

1. **Dual-rate justification**: Can you provide layer-wise gradient magnitudes or forgetting rates showing attention empirically requires faster adaptation?

2. **RMT comparison**: How does RMT with dual-rate EMA perform? How does MiDEA with RMT's replay buffer perform? This isolates true novelty.

3. **Masking loss**: Why cross-entropy? Have you tried $\ell_2$, cosine similarity, or contrastive losses?

4. **Architecture scope**: Does dual-rate EMA work for ViT-L/S, Swin, ConvNeXt, or CNNs?

5. **Stability metrics**: Can you provide explicit forgetting measurements (accuracy on earlier corruptions, distance from source model over time)?

6. **Versatile integration**: Have you explored combining MiDEA's dual-rate EMA with Versatile's sample-selection?

7. **Hyperparameter robustness**: How sensitive is performance to $(\alpha_{\text{attn}}, \alpha_{\text{weight}})$? A 2D heatmap would clarify robustness vs. tuning-dependence.

---

### Official Review · Reviewer_oL71 · 2025-11-01

**Soundness:** 3
**Presentation:** 2
**Contribution:** 1
**Rating:** 4
**Confidence:** 4

**Summary:**

The paper presents **MiDEA**, a decoder-free framework for **continual test-time adaptation (CTTA)** that combines global two-view consistency, local masked self-distillation, and a dual-rate EMA strategy. The method aims to balance adaptation speed and stability while avoiding the computational overhead of multi-pass or replay-based schemes. Empirical results across standard benchmarks (e.g., ImageNet-C, CIFAR-C, VisDA-C) show strong performance, with MiDEA outperforming prior CTTA methods both in accuracy and inference efficiency. The design is particularly tailored to Vision Transformers and achieves high throughput even in batch-1 settings.

**Strengths:**

- **Efficient and lightweight:** Achieves fast adaptation with minimal forward passes (teacher + student) and no replay or ensembling.
- **Strong empirical performance:** Outperforms state-of-the-art CTTA methods on multiple benchmarks with high throughput.
- **Code availability:** The authors provide the code, supporting reproducibility and further research.

**Weaknesses:**

### 1. Limited Novelty and Incremental Design
While the method is well-structured and performs strongly, the overall novelty appears limited. Most components — two-view consistency, patch-level masking, and EMA updates — are established techniques in prior work. Using masking or optimizing different layers at different rates are well-known strategies in vision and self-supervised learning literature. Specifically, the masked modeling used in local self-distillation closely resembles prior work such as Continual-MAE, and the differences in formulation (e.g., use of teacher embeddings vs. decoder) may be implementation-level rather than conceptual innovations. As a result, the method may be better characterized as a careful recombination of existing ideas rather than a fundamentally new contribution.

### 2. Fixed Masking Strategy in Local Distillation
The local masked distillation module uses a fixed, random masking strategy with a constant ratio. This may limit its effectiveness in scenarios with structured or localized corruption, where some regions are more informative or degraded than others. Prior work such as Continual-MAE has explored adaptive masking conditioned on spatial uncertainty or informativeness, which could offer stronger generalization under domain shifts. The lack of adaptivity may cause the model to waste capacity on uninformative patches and reduce the benefits of the masked supervision mechanism.

### 3. No Ablation on Dual-Rate EMA Effectiveness
Although the dual-rate EMA mechanism is one of the paper’s main claimed contributions, the experimental analysis is not sufficient to isolate its true impact. Table 4(b) explores different momentum combinations but does not compare against a well-tuned single-rate EMA baseline. It remains unclear whether a single momentum (e.g., α = 0.95) applied uniformly across all layers would yield comparable performance. More detailed ablations are needed to determine whether the benefit stems from the dual-rate design itself or from tuning effects.

### 4. Lack of Clear Baseline Attribution in Tables
In Tables 1 and 2, the paper compares MiDEA to several prior methods and reports state-of-the-art performance. However, it is often unclear which papers these baselines correspond to, as citations are not explicitly included in the tables themselves. For readers unfamiliar with the abbreviations, this makes it difficult to trace the exact methods or publication venues. It is strongly recommended to include proper citations directly in the tables or captions to improve clarity and attribution. Moreover, Continual-MAE is referred to as an AAAI 2023 work, whereas its official venue appears to be CVPR 2024. Clear and accurate citation of baselines is important for proper attribution and reproducibility.

**Questions:**

Please see Weaknesses.

---

### Note · Authors · 2025-11-13

**Comment:**

We are withdrawing this submission. Thank you for your time and consideration.

**Withdrawal Confirmation:**

I have read and agree with the venue's withdrawal policy on behalf of myself and my co-authors.